# A snow-fire bridge mechanism for the 2025 Southern California winter wildfire

Shizuo Liu [1] ✉, Shineng Hu [1,2] ✉ & Richard Seager [3]

In January 2025, a rare and highly destructive wildfire devastated Southern California, becoming the costliest wildfire event in recorded history. The unusual timing of this wildfire suggests the possibility of unique remote, large-scale climatic precursors that differ from those of previous wildfires. Here, through observational analysis and large-ensemble numerical simulations, we identify that western Eurasian snow cover reduction is associated with weather conditions favorable for wildfires in Southern California in December-January, and simulations indicate that this can occur via an atmospheric teleconnection from western Eurasia, across the North Pacific and into North America. Both observations and simulations show that the reduced snow cover over western Eurasia contributes to the typical wintertime western warming-eastern cooling dipole pattern in North America. The main dynamical mechanisms involve downstream propagating Rossby wave trains triggered by the reduced snow cover in western Eurasia, as well as wave-mean flow interaction over the North Pacific. Our study suggests that Eurasian snow cover anomalies can be predictively linked to both subsequent wildfire risk in California and the wintertime North American zonal dipole temperature pattern, highlighting the broader impacts of Eurasian cryosphere variability on remote climate extremes.

In recent years, extreme wildfires have become more frequent and severe, posing serious threats to natural ecosystems and human society globally[1–4]. Recent examples include the extreme wildfires in Canada in May 2023[5], the devastating fire in Hawaii in August 2023[6], the catastrophic wildfire events in Southern California in January 2025[7,8] and the largest wildfire in South Korea's recorded history in March 2025[9]. Given this trend, there is an urgent need to better understand the climatic drivers behind these events and to raise broader awareness of their impacts.

California has long been a hotspot of wildfires, and global warming has further extended its fire season from summer to autumn or even winter[8,10–12]. In January 2025, a series of extreme wildfires in Southern California killed at least 29 people and caused over 250 billion dollars in damage[7,8], making it one of the most destructive and

costliest events ever recorded in California's fire history. In fact, when a fire occurs in fall or winter in this region, it often causes more damage than those occurring in the summer active-fire season[13] due to enhanced, anomalous atmospheric circulations with stronger winds promoting rapid fire spread[14,15] and typically limited resources like firefighting capacity[11]. Understanding the drivers of winter wildfires in the context of recent climate change is an essential first step toward raising public and governmental risk awareness about these emerging extremes.

Over the past two decades, many studies have examined the climatic drivers influencing the frequency, duration, intensity, long-term trends and interannual variations of summer-autumn wildfires in California. These drivers have been attributed to various local and large-scale factors, including early spring snowmelt[16], reduced

[1]Division of Earth and Climate Sciences, Nicholas School of the Environment, Duke University, Durham, NC, USA. [2]Department of Civil and Environmental Engineering, Duke University, Durham, NC, USA. [3]Lamont Doherty Earth Observatory of Columbia University, Palisades, NY, USA. ✉e-mail: shizuo.liu@duke.edu; shineng.hu@duke.edu

precipitation and humidity[17–19], Santa Ana winds[15,18,20–22], anthropogenic warming[10,12,23–26], and large-scale oceanic and atmospheric variability[24,27–30]. Winter wildfires have rarely occurred in California, where the winter condition is climatologically cold and wet, and thus have received comparatively little attention in scientific research. The extreme wildfires in Southern California in January 2025 and the Thomas Fire in December 2017 challenged this traditional paradigm, pointing to a potentially rising winter fire risk under anthropogenic global warming. Although local meteorological conditions have been shown to contribute to the unusual winter wildfire event this year[8], the potential large-scale climatic drivers behind this event remain poorly understood.

In this study, we aim to investigate the large-scale climatic drivers and the underlying mechanisms of Southern California winter wildfires with implications for the recent 2025 extreme event. In contrast to previous studies that primarily focused on anthropogenic warming and large-scale oceanic and atmospheric variability affecting summer and autumn California wildfires[10,23,24,30], our study has identified a unique but overlooked mechanism for western Eurasian snow cover to influence winter wildfires in Southern California through an atmospheric teleconnection mechanism. Previous observational and modeling studies have shown that extensive and persistent Eurasian snow cover anomalies can induce remote influences[31–35], for example, on the wintertime temperature in eastern North America and western Europe through Rossby wave propagation and the troposphere–stratosphere coupling mechanisms[32,34]. Similarly, other modeling studies have demonstrated that increased autumn snow cover over the Tibetan Plateau can induce Pacific-North America (PNA) or West Pacific (WP)-like atmospheric circulations through eastward propagating Rossby waves and the North Pacific transient eddy feedback, which further influence the North American climate[33,36,37]. Here, extending this view to wildfire extremes, we combine observational analysis and targeted climate model experiments to demonstrate that a preceding reduction of western Eurasian snow cover has potentially contributed to the recent Southern California wildfires in January 2025 through atmospheric teleconnections.

## Results

### Observational hints of the snow-fire teleconnection

Our investigations focus particularly on the December–January (DJ) fire weather index (FWI, see Methods) and vapor pressure deficit (VPD, see Methods). The choice of DJ average, rather than January alone, is motivated by the facts that this particular 2025 wildfire event occurred in early January and that the occurrence of wildfires can be influenced by preceding meteorological conditions[5,38]. Since few wildfires have historically occurred during this season, with the Thomas Fire in December 2017 being the most well-known example, our study uses FWI rather than burned area as the primary metric for establishing the statistical linkage. Previous studies have demonstrated that the FWI corresponds well with observed wildfire events, including remotely sensed fire detections, supporting its use as a proxy for wildfire activity in regions or periods with limited burned area records[23,39]. VPD is another useful metric capturing wildfire potential[40] and is strongly correlated with the FWI in Southern California ($r = 0.92$; Supplementary Fig. 1a). The years witnessing more burned areas correspond closely to the years with extreme FWI and VPD values (Supplementary Fig. 1b–c), although a high FWI or VPD does not necessarily always imply a high wildfire activity that is also influenced by, for example, fuel availability, ignition and fire suppression efforts[23,41].

Wildfires in Southern California like the January 2025 one (Fig. 1c) are typically favored by a local high-pressure system that raises temperature, suppresses precipitation and humidity, and causes Santa Ana winds to penetrate from the inland mountains toward the coas[8,21,42]. These local weather conditions conducive to

wildfires are often thought to be partly driven by large-scale climatic factors especially over oceans[29,30,43]. Among them, La Niña is considered as one of the leading contributors[43,44] and was speculated to have contributed to this unusual wildfire event[45]. However, the La Niña condition in 2024/2025 was rather weak[46]. Our analysis suggests that the anomalously high FWI in December 2024-January 2025 largely exceeded the level that the concurrent La Niña could potentially explain (Supplementary Fig. 2), suggesting the possibility of other large-scale climatic drivers.

We have systematically investigated the statistical relations between the DJ FWI in Southern California and potential climate drivers over global oceans and land. The most robust correlation identified is with the preceding November-December (ND) snow cover extent (SCE) anomalies in western Eurasia (10°E-55°E & 48°N-60°N) on the other side of the globe ($r = -0.59$, $p < 0.01$). Our lead-lag analysis suggests that their correlation maximizes when SCE leads DJ FWI by 1–2 months and drops substantially when SCE lags (Supplementary Fig. 3). Although the El Niño-Southern Oscillation (ENSO) is overall an important oceanic forcing for wintertime climate variability in North America, the correlation between the ND Niño3.4 and the DJ FWI in Southern California is relatively weak ($r = -0.18$). When the influence of ENSO is removed, the snow-fire correlation coefficient is further boosted to -0.64 (Fig. 1a), that is, an anomalously high DJ FWI in Southern California is typically preceded by a reduction in ND SCE in western Eurasia (Fig. 1b, c).

The snow-fire correlation is robust across multiple satellite-derived snow cover datasets and different observational FWI datasets during 2000-2025 (Supplementary Table 1). Detrending does not significantly influence the correlation here as interannual variability dominates over the trend during this period. A nonparametric bootstrap approach is then used to demonstrate that the snow-fire correlation is not dominated by a single extreme year (Supplementary Fig. 4). After constructing a statistical model trained on the western Eurasia ND SCE index over 2000–2024 and then applying it to the hindcast of the 2025 event, we find that it can explain approximately 68% of the VPD anomaly and 40% of the FWI anomaly in 2025 from a statistical perspective.

What is the atmospheric pattern that connects the ND SCE reduction in western Eurasia to the DJ wildfire events in Southern California? The western Eurasian SCE reduction is associated with a pronounced atmospheric wave train, illustrated by the DJ 200 hPa geopotential height and meridional wind fields, that extends from western Eurasia across the continent and the North Pacific to North America, leading to an anomalous anticyclonic circulation over the western U.S. and adjacent seas (Fig. 2a, b). The resultant high-pressure ridge is accompanied by a large-scale subsidence and a reduced cloud cover (Supplementary Fig. 5), which together act to increase surface maximum temperature in the western U.S. through dynamical and radiative warming (Fig. 2c). The strong subsidence also reduces precipitation and relative humidity in this region (Fig. 2d, f), which contributes to the decreased dead fuel moisture and enhanced potential fire intensity over Southern California during December–January (Supplementary Fig. 6). The combined effect of increased temperature and reduced humidity is to significantly enhance the VPD over the western U.S. (Fig. 2g). In addition, the strengthened northeasterly offshore winds near Southern California intensify the Santa Ana winds, which can more rapidly spread wildfires[21,22,47]. All these changes in meteorological conditions contribute to the substantial increase in DJ FWI across the western U.S., especially in Southern California. Such changes in VPD and FWI are already present in November but overall weaker (Supplementary Fig. 7), which can also contribute to subsequent wildfire events. Overall, observational regression analysis suggests that reduced snow cover over western Eurasia may significantly contribute to November-January wildfire

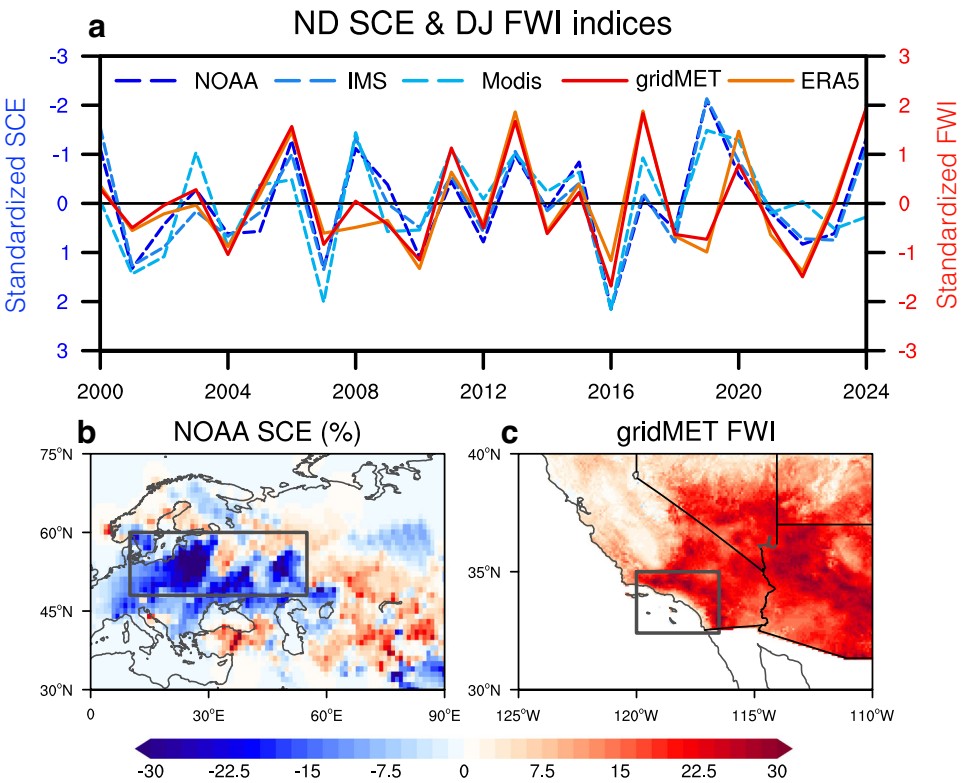

**Fig. 1 | Connections between western Eurasian snow cover extent (SCE) and Southern California fire weather index (FWI). a** Normalized indices with El Niño-Southern Oscillation (ENSO) signal removed for November–December (ND) average western Eurasian SCE for NOAA (the blue dashed line), IMS (the moderate blue dashed line), and MODIS (the light blue dashed line), and for December–January (DJ) average Southern California FWI for gridMET (the red line) and ERA5-based dataset (the orange line) during the period of 2000-2025. The year of DJ is defined by the year of its December. NOAA November-December average western Eurasian SCE (%) anomaly in 2024 (**b**) and gridMET December 2024-January 2025 average Southern California FWI (unitless) anomaly (**c**). The domain boxes in (**b**, 10 °E-55 °E & 48 °N-60 °N) and (**c**, 120 °W-116 °W & 32.5 °N-34.5 °N) are used to calculate the average index for SCE and FWI, respectively.

weather in Southern California through large-scale atmospheric teleconnections.

The SCE reduction in western Eurasia is associated with not only favorable weather conditions for Southern California wildfires but also a continental-scale DJ temperature pattern with warming in western North America and cooling in eastern North America (Fig. 2c). This winter temperature zonal dipole pattern has drawn widespread attention in recent literature[48,49] and has been shown to be related to droughts in the western U.S. and snowstorms in the eastern U.S.[50]. Previous studies have attributed this pattern to various drivers, including atmospheric circulation variability[51,52,53] and sea surface temperature (SST) variations[52]. Our observational analysis suggests that western Eurasian SCE reduction may be another contributing factor to this winter temperature zonal dipole pattern (Fig. 2c; Supplementary Fig. 8) over North America via atmospheric teleconnections (Fig. 2a). This finding lends support to the argument that western Eurasian SCE changes can potentially serve as a useful predictor for winter climate variability across the U.S.[54].

The next key question to explore is whether the snow-fire connection identified in our observational analysis reflects a causal link or merely a simultaneous response to certain atmospheric circulation patterns[55]. It has already been demonstrated in previous modeling studies that Eurasian snow cover changes have the potential to influence the downstream climate through eastward propagating Rossby wave trains along the mid-latitude jet stream[33,56]. Despite that, targeted numerical simulations are necessary to explicitly demonstrate the causality, if any, in the snow-fire connection identified here and to quantify the impact of SCE reduction in western Eurasia on the winter wildfires in Southern California.

## Modeling evidence of the snow-fire teleconnection

To that end, we conduct large-ensemble coupled model experiments with modified SCE conditions in western Eurasia using the Community Earth System Model version 2.1 (CESM2.1)[57] (Methods). Firstly, ten different initial conditions are selected at five-year intervals from a long control simulation. Then, for each initial condition, two sets of 12-member experiments with slightly different atmospheric initial conditions (with a difference of approximately $10^{-14}$ K) are performed with a different prescribed SCE forcing in each set calculated from the NOAA SCE dataset. In the first set (named "ClimSCE"), the climatological SCE from October to January during 2000-2025 is prescribed in western Eurasia (10°E-55°E & 48°N-60°N); in the second set (named "LessSCE"), the true SCE during October 2024-January 2025 is used instead (Supplementary Fig. 9). Snow cover outside this region is computed by the model. All the simulations are integrated from October 1 through February 1 of the following year. Each set consists of 120 simulations in total, and the large-ensemble setup allows us to robustly isolate the signal from the influence of internal climate variability.

In late 2024, the SCE decreased by over 20% of the climatological amount in most of western Eurasia especially after November (Supplementary Fig. 9). The difference between the two sets of simulations (LessSCE minus ClimSCE) highlights the climate response to the 2024/2025 SCE reduction in western Eurasia. The reduced SCE firstly affects the atmosphere regionally by changing surface energy fluxes from November on (Supplementary Fig. 10)[58]. The reduced SCE over western Eurasia lowers surface albedo and increases net downward shortwave radiative flux (Supplementary Fig. 10). The resultant surface warming (Fig. 3a) heats the atmosphere via an enhanced upward longwave radiative flux (Supplementary Fig. 10). The atmospheric

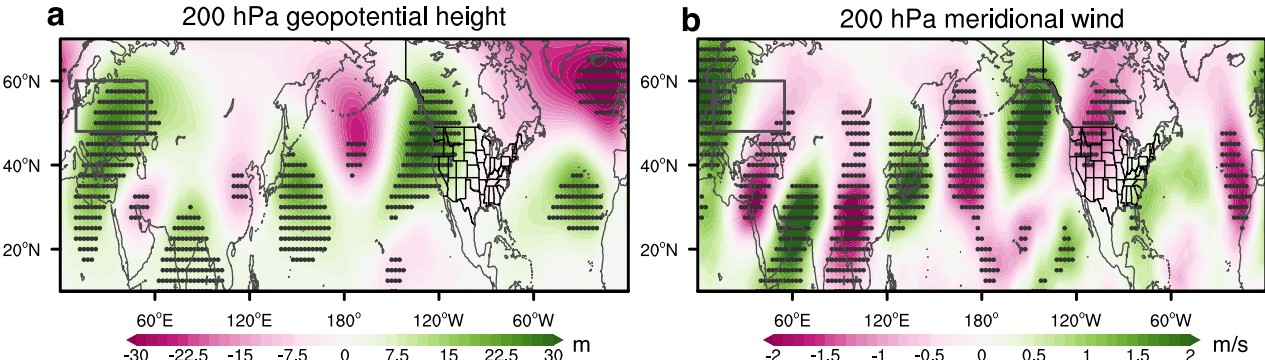

## Observed large-scale atmospheric teleconnection associated with SCE reduction

## Observed local fire-weather condition associated with SCE reduction

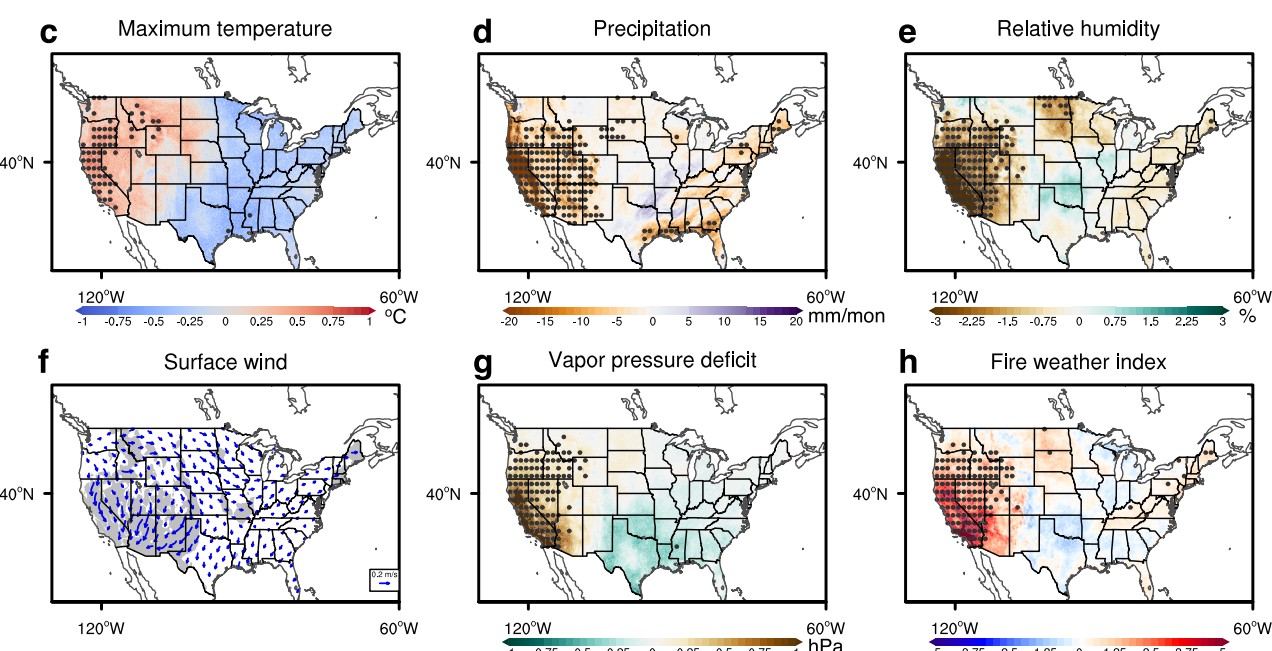

**Fig. 2 | Observed atmospheric teleconnection and U.S. fire-weather condition associated with western Eurasian snow cover extent (SCE) reduction.** Linear regression of December-January average (**a**) 200 hPa geopotential height (m), (**b**) 200 hPa meridional wind (m/s), (**c**) maximum temperature (°C), (**d**) precipitation (mm/month), (**e**) relative humidity (%), (**f**) 10 m surface wind, (**g**) vapor pressure deficit (VPD, hPa), and (**h**) fire weather index (FWI, unitless) against the normalized NOAA November-December average western Eurasian SCE index (the blue dashed line in Fig. 1a, multiplied by −1 to represent reduction) with El Niño-Southern Oscillation (ENSO) removed. Stippling areas in panels (**a**–**e**) and (**g**–**h**), and shading areas in panel (**f**) indicate regions with regression coefficients significant at the 90% confidence level based on the two-sided Student's *t*-test. The black boxes in panels (**a**) and (**b**) represent the domain of western Eurasia (10 °E-55 °E & 48 °N-60 °N).

response has poleward warm advection (Fig. 3b) which, together with the diabatic heating, are balanced by cooling due to adiabatic ascent (Supplementary Fig. 11). The snow cover reduction-induced change in net surface radiative flux (i.e., shortwave and longwave combined) exceeds 25% of its wintertime climatological value.

Such radiative flux changes provide the heat source to trigger the downstream climatic impacts through propagating Rossby waves[33]. Prominent atmospheric wave trains from western Eurasia to North America begin to develop in late October, intensify in November, and stabilize in December (Fig. 3c, d; Supplementary Figs. 12–14). The atmospheric responses consist of two main wave trains, one along the subpolar jet and the other along the subtropical jet (Fig. 3c, d), and they merge over the North Pacific and continue propagating eastward to North America. The simulated atmospheric wave trains overall resemble those in the observed regression pattern (Fig. 2a, b), but mismatches do exist in, for example, the relative weight in the two

wave trains. Those mismatches between the observations and the simulations are not unexpected because the simulations are designed to isolate the atmospheric circulation response to SCE forcing while the observational results may also contain the atmospheric circulation signals that cause the SCE to change.

In response to the SCE reduction, the simulated atmospheric circulation anomalies at the mature phase in DJ are characterized by a North Pacific cyclone and a western U.S. anticyclone (Fig. 3c, d), as in the observed regression patterns (Fig. 2a, b). This atmospheric circulation pattern drives a west-east dipole temperature pattern over North America (or the U.S.) (Fig. 3a), again consistent with observations (Supplementary Fig. 8). Over western North America, the high-pressure system (Fig. 3c) increases daily maximum temperatures (Fig. 3e), reduces relative humidity (Fig. 3g), and thereby enhances VPD (Fig. 3i). The increase in VPD in Southern California is about 33% of that observed in December 2024-January 2025 (i.e., 0.81 hPa in the model

## Simulated large-scale atmospheric teleconnection response to SCE reduction

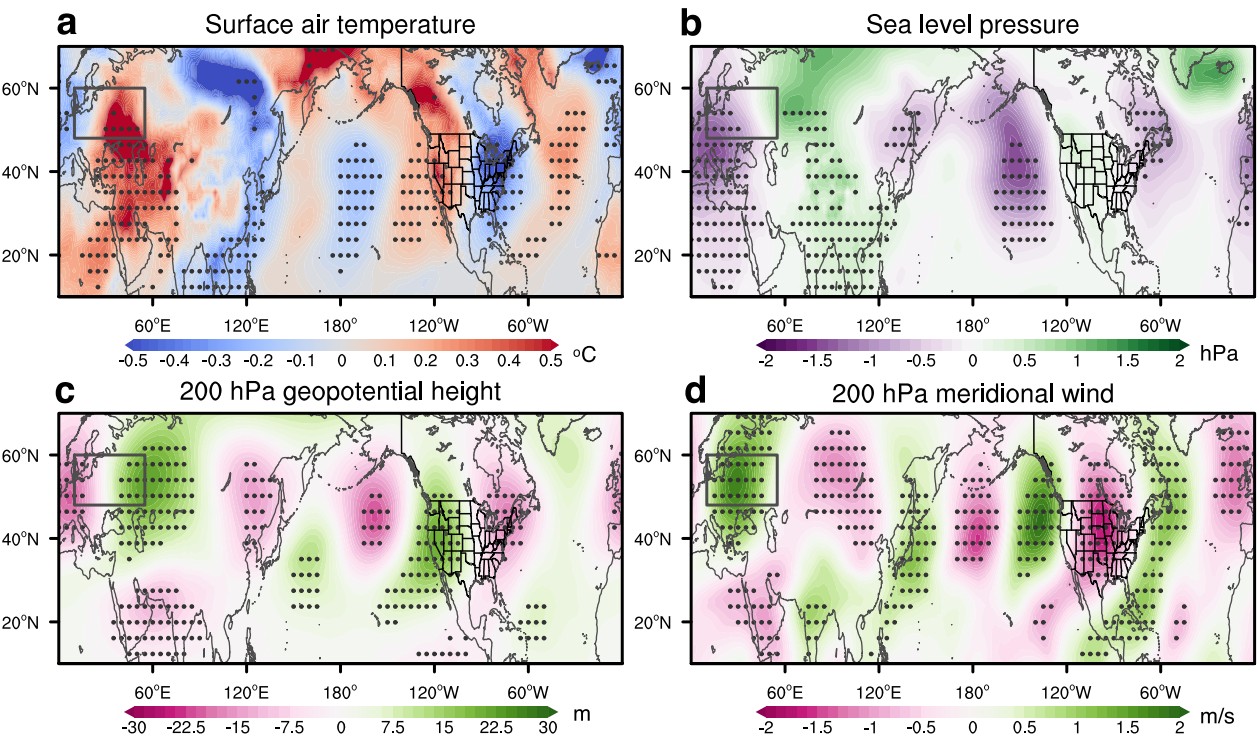

## Simulated local fire-weather condition response to SCE reduction

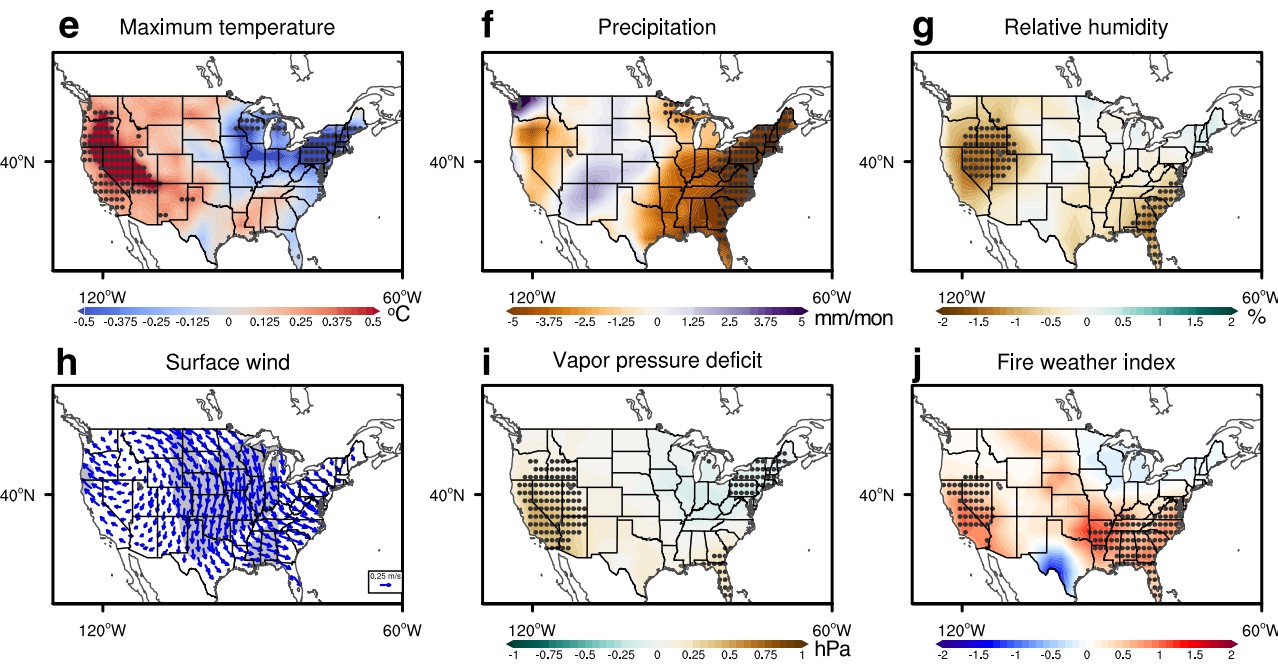

**Fig. 3 | Simulated December-January average atmospheric teleconnection and U.S. fire-weather condition response to western Eurasian snow cover extent (SCE) reduction.** Ensemble mean responses (LessSCE minus ClimSCE) of December-January average (**a**) surface air temperature (°C), (**b**) sea level pressure (hPa), (**c**) 200 hPa geopotential height (m), (**d**) 200 hPa meridional wind (m/s), (**e**) maximum temperature (°C), (**f**) precipitation (mm/month), (**g**) relative humidity (%), (**h**) 10 m surface wind (m/s), (**i**) vapor pressure deficit (VPD, hPa), and (**j**) fire weather index (FWI, unitless). Stippling areas in panels (**a**–**g**) and (**i**, **j**), and shading areas in panel (**f**) indicate responses significant at the 90% confidence level based on the two-sided Student's t-test. The black boxes in panels (**a**–**d**) represent the domain of western Eurasia (10 °E-55 °E & 48 °N-60 °N).

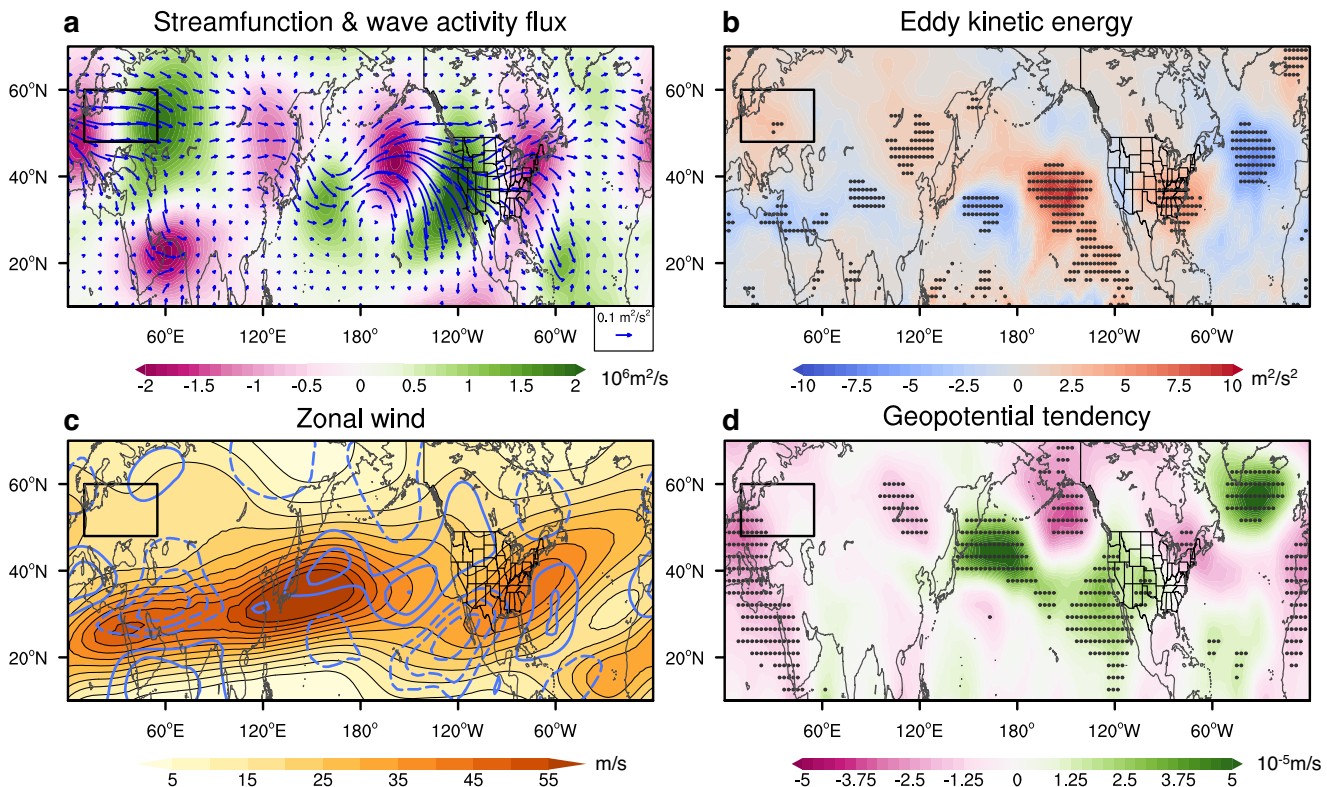

**Fig. 4 | Dynamical diagnostics of the 200 hPa atmospheric circulation response.** Ensemble mean responses (LessSCE minus ClimSCE) of December-January average 200 hPa (**a**) streamfunction (shading, $10^6$ m$^2$/s) and stationary horizontal wave activity flux (WAF, vectors), (**b**) eddy kinetic energy (EKE, m$^2$/s$^2$), (**c**) zonal wind (shown as the blue contours, m/s), (**d**) geopotential tendency ($10^{-5}$ m/s). In (**c**), the contour interval is 0.6 m/s, with the blue solid (gray dashed) contours denoting positive (negative) values, and zero contours are omitted; shading denotes the CESM2.1 long-term climatological 200 hPa zonal wind during December–January (m/s). Stippling areas in panels (**b**) and (**d**) indicate responses significant at the 90% confidence level based on the two-sided Student's t-test. The black boxes in panels (**a**–**d**) represent the domain of western Eurasia (10 °E-55 °E & 48 °N-60 °N).

vs. 2.48 hPa in the gridMET dataset). In Southern California, although the strengthened southeasterlies (Fig. 3h) can intensify the Santa Ana winds, their magnitude is weaker than observed due possibly to the model limitation in accurately resolving the topography and simulating wind[59]. Additionally, the simulated precipitation reduction in Southern California (Fig. 3f) is notably weaker than in observations. As a result, the magnitude of the simulated FWI increase (Fig. 3j) is also weaker than observed. These results suggest that the western Eurasian SCE reduction can be one contributor to January wildfires in Southern California through VPD-related large-scale atmospheric circulation changes. The SCE impact on local precipitation and surface winds would be well worth studying in models that have a better representation of regional orographic features.

### Dynamic mechanisms of the snow-fire teleconnection

Next, dynamic diagnostics are used to investigate the underlying mechanisms for the SCE reduction in western Eurasia to influence the wintertime large-scale remote atmospheric circulation response associated with Southern California wildfires and the North American dipole temperature anomaly. As previously discussed, the sustained reduction in snow cover over western Eurasia leads to enhanced upward wave activity flux (WAF), indicating vertical propagation of Rossby wave energy into the mid-to-upper troposphere (Supplementary Fig. 15), and is associated with the formation of a prominent Rossby wave source[60] on its eastern flank (Fig. 4a). The simulated 200 hPa Rossby waves primarily propagate downstream, as shown by the horizontal WAF along the subpolar jet (Fig. 4c), with part of the wave energy refracted southeastward toward the subtropical jet. Then, the Rossby waves further propagate eastward along the North Pacific

midlatitude jet[61], intensify at the North Pacific midlatitude jet exit, and continue eastward to North America and Europe (Fig. 4a; Supplementary Fig. 15).

In DJ, the strong WAF in the North Pacific midlatitude jet exit (Fig. 4a, c) seems to imply a role of wave-mean flow interaction[62]. Over the midlatitude North Pacific storm track region, the eddy kinetic energy (EKE) response exhibits a marked increase (Fig. 4b). The impacts of the synoptic eddies on the mean flow at the 200 hPa level are then calculated via the quasi-geostrophic potential vorticity (QGPV) equation[63] (Methods). Changes in transient eddies induce a decrease in 200 hPa geopotential height over the central North Pacific and an increase in geopotential height on its eastern flank, consistent with the simulated 200 hPa (Fig. 3c) geopotential height responses. These results show that, in addition to the eastward-propagating Rossby waves, atmospheric feedbacks involving synoptic-scale transient eddies are also critical to the formation of the circulation patterns over the North Pacific-North America sector in response to the SCE reduction[33]. It is noteworthy that the atmospheric circulation response is comparatively weaker in November because the subtropical and the subpolar jets are considerably weaker and the wave-mean flow interaction is weaker[64] (Supplementary Figs. 16 and 17). Also, the North Pacific midlatitude jet exit is displaced farther north in November than in DJ (Supplementary Fig. 16), limiting its influence in the western U.S.

## Discussion

California has historically experienced frequent wildfires, but their occurrence has recently extended beyond the traditional summer months into year-round events[10]. The urban wildfires in Southern California this January caused unprecedented economic losses,

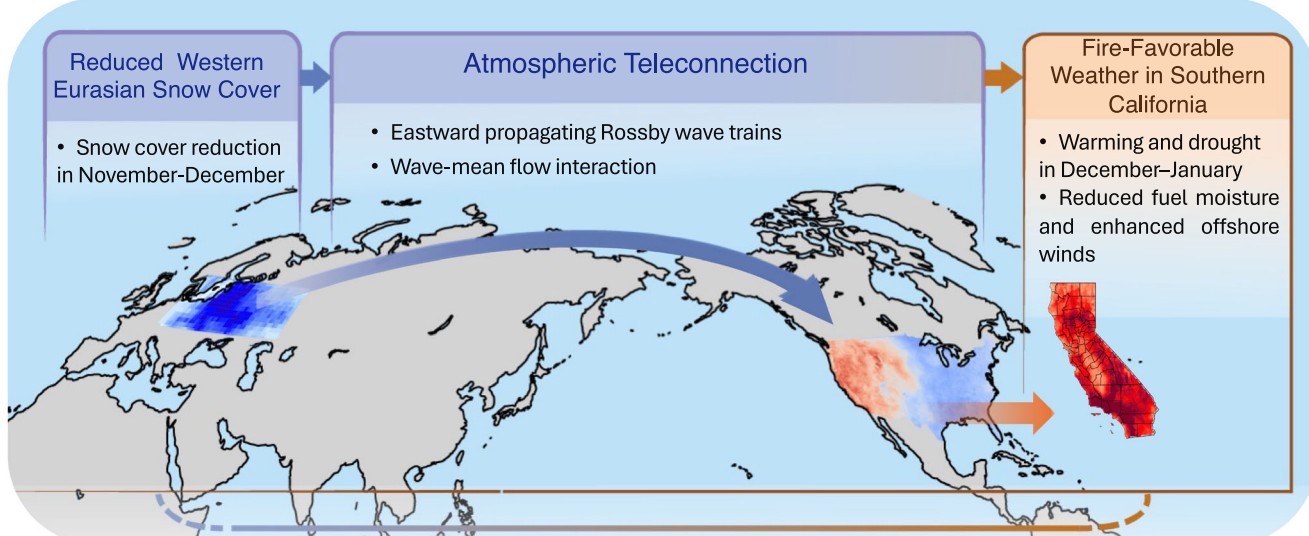

**Fig. 5 | Schematic illustration of the snow–fire teleconnection mechanism.** Schematic diagram illustrating the proposed teleconnection linking reduced western Eurasian snow cover in early winter (November–December) to fire-favorable weather conditions in Southern California during December–January. Reduced snow cover over western Eurasia excites Rossby wave trains that establish large-scale atmospheric teleconnections, favoring the development of a persistent high-pressure system over the western U.S. with a typical wintertime western warming-eastern cooling dipole pattern across the continental U.S. This circulation results in warmer and drier conditions, reduced fuel moisture, and enhanced offshore winds in Southern California, thereby increasing winter wildfire risk.

highlighting an urgent need to investigate the mechanisms driving such unusual fire activity. Motivated by this question, we propose a mechanism wherein atmospheric circulation anomalies triggered by western Eurasian snow cover variability connects antecedent SCE anomalies with the occurrence of wildfires in Southern California in January (Fig. 5). Our study, combining observational analysis and model simulations, indicates that the SCE reduction since autumn over western Eurasia can perturb the local surface energy balance, trigger an eastward-propagating atmospheric wave train, and then induce an anomalous anticyclonic circulation over western North America during December-January. This anticyclonic circulation anomaly promotes clearer skies, reduced cloud cover, higher daily maximum temperatures and lower relative humidity across the western U.S., ultimately enhancing the VPD and FWI, especially in Southern California. Based on our simulation-based estimate, the large-scale atmospheric response to the western Eurasian snow cover reduction alone accounts for about one third of the VPD increase associated with the 2025 extreme wildfire. It remains unclear to what extent the SCE reduction can influence the Southern California wildfires through mesoscale precipitation and local Santa Ana wind events, which are often poorly represented in global climate models[59]. It would be well worth studying this in high-resolution models that have a better representation of regional orographic features in the future.

Further analysis of model simulations indicates that upstream snow cover anomalies can influence the winter atmospheric circulation over western North America through the eastward propagation of Rossby waves originating from the snow forcing region along the subpolar jet and subtropical jet, coupled with interactions between synoptic-scale transient eddies and the zonal mean flow within the North Pacific storm track region[63]. This Rossby wave propagation in mid-high latitudes over the North Pacific largely reflects the typical mechanism through which large-scale climate variability in upstream regions influences winter climate over the western U.S.[55,65]. However, very few previous studies have explicitly considered the active role of snow cover anomalies. Both the observational and modeling results from our study consistently reveal that the reduced SCE over western Eurasia can induce a large-scale east–west dipole in atmospheric circulation over North America, leading to a corresponding dipole

temperature pattern characterized by warming in the west and cooling in the east. This recurrent dipole temperature pattern has attracted considerable attention due to its frequent occurrence and broad impacts[48,52,53]. Our study identifies SCE anomalies over western Eurasia as a previously underrecognized contributor to its development.

Recently, many studies have focused on the influence of local meteorological factors and fuel conditions on wildfires in the western U.S.[14], while relatively few have examined the role of remote factors, most of which concern the impact of SST[19,38]. To fill this gap, we demonstrate that SCE anomalies over the mid-high latitudes of Eurasia can influence California wildfires through a typical midlatitude atmospheric teleconnection mechanism. Our results indicate that studies of the dynamic mechanisms governing North American atmospheric circulation should incorporate upstream Eurasian hydrological processes. Our results further highlight the potential of incorporating Eurasian snow cover as a predictor of California wildfires along with other predictors, consistent with earlier studies that have already shown that Eurasian autumn snow cover can serve as an effective predictor of winter temperatures in eastern North America[54]. The PNA and the Arctic Oscillation (AO) are two important atmospheric patterns that could influence the winter climate variability in North America. Indeed, both the PNA and the AO indices lead the DJ FWI in Southern California by 1-2 months, but their correlations are relatively weak compared to that of western Eurasian SCE (Supplementary Fig. 18). Our multivariate linear regression analysis suggests that further including October-November PNA and AO as predictors, in addition to western Eurasia ND SCE, increases the explained variance of Southern California FWI index from 35% to 49%. It implies that western Eurasia ND SCE is a dominant contributor to the winter wildfire weather conditions, while other internal variability like PNA and AO can also contribute.

Our study mainly focuses on the impacts of snow cover anomalies, rather than their underlying causes. The observed reduction of western Eurasian SCE may be influenced by multiple factors, including dominant circulation modes in the North Atlantic–European sector (e.g., the North Atlantic Oscillation or the East Atlantic), North Atlantic SST, and Arctic sea ice variability. However, our preliminary analyses suggest that these factors do not seem to be directly linked to either

the western Eurasian SCE or winter FWI in California. The mechanisms controlling the year-to-year snow cover variability and its associated atmospheric circulation pattern remain an open question.

The snow-fire teleconnection mechanism proposed in this study may have potential implications for assessing the future changes of wildfire activity in a warmer climate. First, although this study is focused on interannual variability, the snow-fire teleconnection mechanism should also operate for externally forced climate changes. Whether the snow melt in western Eurasia under a warming climate will enhance winter wildfire activity in California awaits to be confirmed. Second, as both SCE variability and atmospheric jets may alter under a warming climate with a rapid warming of the Arctic, the snow–atmosphere teleconnection and its impact on California wildfire activity may be non-stationary[34]. We attempt to explore this by investigating the snow-fire correlations in 30 pairs of historical and future simulations from 8 climate models in the Coupled Model Intercomparison Project Phase 6 (CMIP6). Only a limited number of historical simulations exhibit statistically significant correlations, and none of them can reach the correlation seen in observations, although models reasonably simulate the magnitude of SCE variability itself (Supplementary Fig. 19). This is consistent with previous studies showing that coupled climate models often struggle to internally capture key snow–atmosphere teleconnections[34,66,67]. The model failure in reproducing the observed snow–fire linkage prevents us from drawing solid conclusions on how the snow-fire teleconnection may possibly change under a warming climate. Finally, fire–fuel nonlinearities may further increase the complexity of the future snow–fire relationship changes, as future warming can push fuel dryness across critical thresholds, triggering abrupt surges in fire activity[68]. Future studies are needed along those lines.

## Methods
### Observational datasets
In this study, we employ two Canadian Fire Weather Index (FWI) datasets to represent potential fire danger[69], which are widely used in western U.S. wildfire research[29,30]. These two FWI datasets are gridMET FWI[70] and ERA5-based FWI[71]. The gridMET FWI provides daily high-spatial resolution (~4 km, 1/24th degree) data over the contiguous U.S. from 1979 to the present. The ERA5-based FWI, produced by the Copernicus Emergency Management Service, is derived from ERA5 reanalysis data with a spatial resolution of 0.25° and daily temporal resolution, spanning from 1 January 1979 to the present. Detailed descriptions of these two datasets are available in ref. 30. Given the historically low occurrence of wildfires in California during January, using the FWI is more suitable for statistical analysis than relying on the limited in-situ fire records. The correlation between gridMET FWI and ERA5-based FWI is presented in Supplementary Table 1. In addition, we use other meteorological variables from gridMET[70] and ERA5[72] in our analysis. In addition, we also use a new observational database of locations and sizes of individual fires in the western U.S. [Western U.S. MTBS-Interagency (WUMI) wildfire database][73]. It includes wildfire events in the western U.S. that are ≥100 hectares (1 km²) in size from 1984 to the present, provided by a number of government sources.

Three Northern Hemisphere snow cover extent (SCE) satellite datasets are used in this study. The first dataset is the NOAA snow chart climate data record (CDR), archived and managed by the Rutgers University Global Snow Lab[74]. This dataset combines weekly (1966–May 1999) and daily (June 1999–Present) NOAA visible satellite-based SCE analyses to form a continuous record of snow coverage for the entire Northern Hemisphere, with a 24 km resolution. The second dataset is the Interactive Multisensor Snow and Ice Mapping System (IMS) snow cover data[75], which covers February 1997 to present and is derived from a variety of data products, including satellite imagery and in situ data. This dataset provides daily SCE at a 24-km resolution. The third dataset is the MODIS monthly snow cover[76] covering March 2000

to the present, with a spatial resolution of 0.05° x 0.05°. In this study, we analyze snow cover from 2000 to the present, which represents the period of overlap among the three datasets with relatively higher data accuracy[77]. The strong correlations among the three datasets during 2000–2024 demonstrate their overall consistency and reliability (Supplementary Table 1). We mainly use the NOAA CDR SCE dataset for analysis and as the prescribed snow forcing in model simulations.

We also use NASA Goddard's Global Surface Temperature Analysis (GISTEMP)[78], which combines land surface air temperatures from GHCN-M version 4 with SSTs of the NOAA Extended Reconstructed SST Version 5 (ERSSTv5)[79] into a comprehensive global surface temperature dataset. It spans 1880 to the present at monthly resolution, on a 2° x 2° latitude-longitude grid. In this study, the Niño3.4 index is defined by convention as ERSSTv5 SST anomalies averaged within 5°S–5°N and 170°W–120°W. Prior to further analysis, a partial regression is performed to remove the influence of ENSO from the observational datasets using Gram–Schmidt orthogonalization[80].

### CMIP6 historical and future scenario runs
Monthly data from 13 CMIP6 models (Supplementary Table 2) are used in this research. We use all models with available monthly FWI data[81]. In particular, we use each model's available 'r1i1p1f1', 'r2i1p1f1', 'r3i1p1f1', 'r4i1p1f1' and 'r5i1p1f1' ensemble members, from two experiments: historical and SSP5-8.5. In total, 44 simulation runs are included in the analysis, each with FWI data for both the historical (1850–2014) and SSP5-8.5 (2015–2100) experiments.

### Model simulations
To further validate the phenomena observed, we conduct a series of snow cover modification experiments with the Community Earth System Model 2.1 (CESM2.1)[57]. The atmospheric and land components use the f09 finite-volume grid (0.9° latitude × 1.25° longitude), while the ocean and sea-ice components use the gx1v6 displaced pole grid with a nominal resolution of ~1° × 1°. Our earlier studies investigating the climate effects of Eurasian SCE with an earlier CESM version demonstrated the model's robustness, supporting its application in the present study[33,37]. In this study, the SCE forcing is prescribed over western Eurasia (10 °E-55 °E & 48 °N-60 °N) based on the NOAA snow chart CDR snow cover dataset, and model derived SCE is used everywhere else.

First, ten different initial conditions are selected at five-year intervals from a 100-year long control simulation. Then, for each initial condition, two sets of 12-member experiments with slightly perturbed atmospheric initial conditions that differ by $10^{-14}$ K[82] are performed. Each run is integrated from October 1 through February 1 of the following year. The control set ("ClimSCE") uses the climatological October–January SCE averaged over 2000–2025 as snow forcing, while the forced set ("LessSCE") uses the observed SCE from October 2024 to January 2025. Thus, each ensemble begins with identical initial conditions and gradually diverges due to the imposed differences in snow boundary conditions. Each set consists of 120 simulations, and the response is defined as the difference between the ensemble means of the LessSCE and ClimSCE runs (LessSCE minus ClimSCE). In the model experiments, SCE forcing is prescribed from October through January to ensure a sustained and robust forcing, thereby facilitating a significant winter atmospheric circulation response[33,37].

The land model used in this study is CLM5.0[83], which outperforms earlier versions (i.e., CLM4.5 and CLM4.0) in many aspects, including its enhanced ability to simulate snow-related hydrological processes. At each model time step, snowfall is permitted to accumulate at the surface. Melting of existing snow cover absorbs latent heat from the atmosphere and adds soil moisture to the land model. In our simulation, CLM5.0 is modified such that the simulated SCE over western

Eurasia is replaced at each time step with a prescribed SCE for the specified snow season, interpolated from monthly values. As a result of this modification, energy and water are not explicitly conserved within the western Eurasia region in each ensemble[84].

In this study, the idealized snow experiments were conducted with the fully coupled CESM2.1 model rather than an atmosphere-only (AGCM) model. While the model is fully coupled, our interpretation focuses on the atmospheric circulation response to the imposed western Eurasian snow cover anomalies, because the associated North Pacific SST response is quite weak (<0.1 °C) (Supplementary Fig. 20a), and therefore unlikely to exert substantial feedback on the winter atmospheric circulation through North Pacific ocean–atmosphere coupling. Besides, the response of Arctic SIC is also negligible (Supplementary Fig. 20b).

### Vapor pressure deficit (VPD)
Vapor pressure deficit (VPD) represents the difference between the actual vapor pressure and the saturation vapor pressure, reflecting the drying power of the air. It is calculated from daily maximum surface air temperature ($T_{max}$, K) and surface relative humidity (RH, %) as follows[30]:

$$e_s = 611.2 * \exp(17.67 * \frac{T_{max} - 273.15}{T_{max} - 29.65}) \quad (1)$$

$$VPD = e_s(1 - RH) \quad (2)$$

where $e_s$ is the saturated vapor pressure in Pa.

### Fire weather index (FWI)
Following the methodology used in the gridMET and ERA5 FWI products, we calculate the daily FWI response in the snow cover forcing experiments using the Canadian FWI system as described by ref. 69. To calculate the FWI, we use the daily maximum surface air temperature ($T_{max}$, K), surface relative humidity (RH, %), precipitation (Pr, mm/day) and surface wind speed (WS, m/s).

The FWI system comprises several indices and is calculated in three steps[69,81]. The first step contains computing three fuel moisture codes: Fine Fuel Moisture Code (FFMC), Duff Moisture Code (DMC), and Drought Code (DC), which represent the moisture content of different forest fuel layers. In the second step, the Initial Spread Index (ISI) and Buildup Index (BUI) are derived from the fuel moisture codes and surface wind speed to estimate the potential rate of fire spread and the available fuel for combustion. Finally, the Fire Weather Index (FWI) is computed by combining the ISI and BUI to produce a numerical rating of potential fire intensity. The FWI system is driven exclusively by meteorological inputs and does not account for factors such as vegetation type, ignition sources, or fire suppression efforts. For further method details, refer to ref. 69 or ref. 81.

### Rossby wave activity flux (WAF)
WAF is used to study the propagation of Rossby waves. In this study, we apply the WAF formula proposed by ref. 85 as a dynamical diagnostic tool for the three-dimensional propagation of Rossby waves. The horizontal component of the WAF, based on the monthly mean 200 hPa geostrophic wind, can be expressed as follows:

$$W = \frac{p}{2|\boldsymbol{U}|} \begin{bmatrix} U(\psi'^2_x - \psi'\psi'_{xx}) + V(\psi'_x\psi'_y - \psi'\psi'_{xy}) \\ U(\psi'_x\psi'_y - \psi'\psi'_{xy}) + V(\psi'^2_y - \psi'\psi'_{yy}) \end{bmatrix} \quad (3)$$

where $\boldsymbol{U}(U, V)$ is climatological horizontal wind during 2000-2024, $\psi'$ is the monthly streamfunction anomaly derived from the 200 hPa geopotential height by the quasi-geostrophic approximation method,

and $x$ and $y$ represent the longitude and latitude directions, respectively. The pressure, $p$, is standardized by 1000 hPa.

### Horizontal eddy kinetic energy (EKE)
The horizontal EKE is calculated based on the perturbations of the 200 hPa horizontal wind speed[63], and primarily reflects storm track changes:

$$EKE = \overline{(u''^2 + v''^2)/2} \quad (4)$$

where $u''$ and $v''$ are zonal and meridional wind components derived from daily data with 2–8 day band-pass filtered fluctuations, respectively. The overbar denotes the monthly mean, and double-prime represents deviations of daily values from the corresponding long-term daily climatological mean.

### Geopotential height tendency $\partial z/\partial t$
The dynamical feedback of synoptic eddy activities onto 200 hPa time mean flow can be quantitatively calculated using the two-dimensional quasi-geostrophic potential vorticity (QGPV) equation[63]. The geopotential height tendency $\partial z/\partial t$ due to transient eddies is proportional to the inverse Laplacian of the convergence of the vorticity flux of transient eddies ($\pi$):

$$\frac{\partial z}{\partial t} = \frac{f}{g}\nabla^{-2}\pi \quad (5)$$

where

$$\pi \equiv \frac{1}{a^2\cos\theta}\left(\frac{\partial}{\partial\theta}\frac{1}{\cos\theta}\frac{\partial}{\partial\theta}\cos^2\theta - \frac{1}{\cos\theta}\frac{\partial^2}{\partial\lambda^2}\right)\overline{u''v''} + \frac{1}{a^2\cos^2\theta}\frac{\partial^2}{\partial\lambda\partial\theta}\cos\theta\left(\overline{u''^2} - \overline{v''^2}\right) \quad (6)$$

$a$ is Earth's radius, $f$ is the Coriolis parameter, $g$ is the gravitational acceleration, $\lambda$ is longitude, and $\theta$ is latitude. The geopotential height tendency equation is used to evaluate the effect of transient eddies on geopotential height. $u''$ and $v''$ are zonal and meridional wind components derived from daily data with 2–8 day band-pass filtered fluctuations. The overbar denotes the monthly mean, and double-prime represents deviations of daily values from the corresponding long-term daily climatological mean.

## Data availability
The data used in the manuscript are publicly available for gridMET (https://www.northwestknowledge.net/metdata/data/), WUMI burned area (http://datadryad.org/share/Ox4oxdwdrhkmjUTpke7QgkfF--h-RLRbmMzGBhSmOr4), ERA5-based FWI (https://cds.climate.copernicus.eu/cdsapp#!/dataset/cems-fire-historical?tab =form), ERA5 (https://cds.climate.copernicus.eu/cdsapp#!/dataset/reanalysis-era5-single-levels-monthly-means?tab=form and https://cds.climate.copernicus.eu/cdsapp#!/dataset/reanalysis-era5-pressure-levels-monthly-means?tab=form), NOAA SCE (https://www.ncei.noaa.gov/data/snow-cover-extent/access/), IMS SCE (https://noaadata.apps.nsidc.org/NOAA/G02156/), MODIS SCE (https://nsidc.org/data/data-access-tool/MOD10CM/versions/61), GISTEMP, CMIP6, CMIP6 FWI (https://www.research-collection.ethz.ch/entities/researchdata/1b12cf29-794a-4987-b455-6164ba5c4294), snow cover forcing experiments (https://zenodo.org/records/17554730).

## Code availability
The current CESM2.1 version is freely available at www.cesm.ucar.edu/models/cesm2/. The NCL scripts used to generate the main plots in this paper are available from https://doi.org/10.24433/CO.4610641.v1.

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

## Acknowledgements

This work was supported by NASA grants 80NSSC22K1025 (S.L., S.H.) and 80NSSC23K0988 (S.H.); the DOE grant DE-SC0024186 (S.H.); and the NSF grant AGS-2127684 (R.S.).

## Author contributions

S.L. conceived the original idea, performed the data analysis and numerical simulations and, together with S.H. and R.S., designed the research, interpreted the results, and wrote the manuscript.

## Competing interests

The authors declare no competing interests.
