## [Transparent Peer Review file · Nature Communications]

A Snow-Fire Bridge Mechanism for the 2025 Southern California Winter Wildfire

Corresponding Author: Dr Shizuo Liu

Version 0:

Reviewer comments:

Reviewer #1

(Remarks to the Author)

The manuscript presents an original and scientifically compelling framework linking western Eurasian snow cover anomalies to winter wildfire risk in Southern California. The observational–model synthesis is strong; however, the causal attribution could be strengthened by explicitly quantifying the contribution of competing drivers (e.g., ENSO, PDO, Arctic sea-ice anomalies) rather than describing them qualitatively. A sensitivity analysis or multi-model comparison section would help position the proposed mechanism relative to established teleconnections.

Major comments

1. The CMIP6-based future risk interpretation remains somewhat speculative. The manuscript currently implies future predictability based on the mechanistic link, but the robustness of this signal under warming—given snow regime shifts, fire–fuel nonlinearities, and model spread—is not fully assessed. A short uncertainty framing (even if exploratory) would make the conclusions more aligned with Nature Communications standards.
2. Validation of the fire weather signal would benefit from incorporating independent fire danger metrics beyond FWI and VPD (e.g., fuel moisture datasets, burned area residual analysis, or regional fire-weather classifications), particularly because winter fires are rare and observational record length is short.
3. The authors should cite relevant work in which statistically derived fire weather indices have been compared against remotely sensed fire occurrence. In particular, Dhanurkar et al. (2024) demonstrated a strong spatiotemporal match between the Canadian Fire Weather Index and MODIS fire detections, which supports the methodological use of FWI as a proxy in regions or periods where burned area records are sparse. Including this strengthens the justification for using FWI in a low-fire-season context.

Minor comments

1. Figures 2 and 3 are dense and could benefit from clearer labeling and a more explicit visual guide to the causal sequence (e.g., schematic or pathway figure).
2. Please clarify whether detrending procedures were applied symmetrically across predictor and response variables, and whether the correlation strength is sensitive to the detrending method.
3. Some terminology (e.g., “bridge”, “pathway”) reads conceptual rather than mechanistic; consider aligning phrasing more closely with accepted atmospheric teleconnection nomenclature.
4. For reproducibility, the exact boundaries of the western Eurasian domain used in each analysis should be stated consistently in both main text and figures (some places define it only in captions).
5. Minor typographical and formatting edits needed, particularly in multi-panel figure captions and units formatting.

Reference

Dhanurkar, T., Budamala, V., & Bhowmik, R. D. (2024). Understanding the association between global forest fire products and hydrometeorological variables. *Science of The Total Environment*, 945, 173911.

(Remarks on code availability)

I checked the codes which are mainly plotting codes.

Reviewer #2

(Remarks to the Author)

This study links the record-breaking winter 2025 wildfires in southern California to anomalously low snow cover over western Europe and proposes a “snow–fire bridge” teleconnection: reduced European snow triggers a downstream Rossby-wave train that is enhanced by wave–mean-flow interaction over the North Pacific, thereby creating meteorological conditions conducive to fire in southern California. The work provides a novel, physically based global perspective on the remote drivers of such extreme events. The manuscript is generally well organised and the conclusions are robust, but the following points need clarification or additional analyses to reach Nature Communications’ standards.

Comments

- 1.This study used the FWI instead of burned area to investigate the remote drivers for the wildfires in South California. As shown in Fig. S3, the high FWI may provide a favorable condition for wildfires as in 2017 and 2024, but its impact is not necessarily inevitable, such as in 2011 and 2013. This should be clarified in the manuscript.
- 2.Provide correlation coefficients (and significance levels) between western European snow anomalies and southern California winter FWI/burned area for 1920–2024 after excluding 2025, ideally with Monte-Carlo or bootstrap tests, to demonstrate that the teleconnection is not dominated by a single extreme year.
- 3.Snow-cover anomalies are surface-based, yet the key wave response is shown in the mid-to-upper troposphere. Please add a longitude–height cross-section of wave-activity flux (WAF or EP-flux) along the wave path to demonstrate how the surface anomaly initiates upward-propagating Rossby waves. A brief dynamical discussion should also be included.
- 4.The manuscript promotes Eurasian snow as a predictor. Quantify this: build a statistical model trained on 2001-2024 snow-cover indices and hindcast the 2025 event. Report cross-validated skill scores and the amplitude of the predicted versus observed Southern California VPD/FWI. If skill is modest, state this clearly.
- 5.The analysis assumes the relationship is stationary over the observational period. Given climate change and the rapid warming of the Arctic, the teleconnection pathway’s strength and structure may be non-stationary. This should be discussed as a key uncertainty and limitation.
- 6.European snow anomalies explain only ~one-third of the local vapour-pressure deficit (VPD). Please discuss: What drives the remaining VPD increment (e.g., local subsidence warming, advection of dry/warm air, SST anomalies)? In addition, why 2017 exhibited high FWI and large burned area despite weak snow anomalies, to delineate the boundary conditions of the snow–fire bridge.
- 7.The idealized snow experiments must be clearly detailed. Were they atmosphere-only (AGCM) runs with prescribed SSTs and snow? Or coupled? This has major implications for the interpretation.
- 8.Unit is missing in Fig. S8

(Remarks on code availability)

Version 1:

Reviewer comments:

Reviewer #1

(Remarks to the Author)

I have carefully examined the revised manuscript and the authors’ detailed point-by-point responses. I appreciate the substantial effort invested in addressing the concerns raised during the previous review. The additional analyses, clarifications, and expanded discussion have significantly strengthened the robustness, transparency, and overall presentation of the study.

In particular, the inclusion of bootstrap testing, independent fire danger metrics, the statistical hindcast experiment, and the expanded discussion on uncertainties and non-stationarity have addressed my main concerns. The revisions are thorough and appropriate.

I am satisfied with the revision and support the manuscript for publication in its current form.

(Remarks on code availability)

Reviewer #2

(Remarks to the Author)

My concerns have been satisfactorily addressed. I have no further comments.

(Remarks on code availability)

I was able to run the codes and reproduce the results. To make this more useful for the community, it would be helpful if the author could package the main logic into a reusable function.

Responses to the reviews of the manuscript

“A Snow-Fire Bridge Mechanism for 2025 Southern California Winter Wildfire”

Shizuo Liu, Shineng Hu, Richard Seager

We thank the reviewers for their time evaluating our work and for providing both encouraging feedback and constructive suggestions. Their insightful comments are vital in improving the clarity and rigor of our key findings. We have carefully addressed the reviewers' suggestions, and the revised manuscript includes the following key changes:

1. We examined and compared the lead-lag correlations between the Southern California December–January (DJ) FWI and different climate indices. We further constructed a multivariate linear regression model using major indices as predictors. These results indicate that, although some climate modes may contribute to DJ fire weather conditions in Southern California, western Eurasian SCE emerges as the dominant contributor (as suggested by Reviewers 1 and 2).
2. We expanded the discussion on the implications of the snow–fire teleconnection for future wildfire activity under a warmer climate. Specifically, we discussed the applicability of this mechanism to externally forced climate changes, examined snow–fire correlations in CMIP6 simulations for both historical and future periods, and noted that fire–fuel nonlinearities may further complicate the future snow–fire relationship (as suggested by Reviewers 1 and 2).
3. We clarified the interpretation and appropriate use of FWI as a fire danger indicator to ensure a more accurate understanding in the context of actual wildfire events (as suggested by Reviewers 1 and 2).
4. We applied bootstrap tests to reinforce the robustness of our results and to demonstrate that the snow–fire relationship is not dominated by a single extreme year (as suggested by Reviewer 2).
5. We constructed a statistical model based on the western Eurasian November–December SCE index (2000–2024) and applied it to hindcast the 2025 event. The model explains approximately 68% of the 2025 VPD anomaly and 40% of the FWI anomaly from a statistical perspective (as suggested by Reviewer 2).
6. We incorporated additional independent fire danger metrics, including dead fuel moisture and the energy release component, to further strengthen the robustness of our main results (as suggested by Reviewer 1).
7. We have included a schematic diagram, to better illustrate the dynamical processes of the snow–fire teleconnection proposed in our study (as suggested by Reviewer 1).

8. We provided a more detailed description of the fully coupled CESM2.1 experiments used in this study and clarified our focus on the atmospheric pathway, given the quite weak oceanic responses (as suggested by Reviewer 2).

9. We improved the figure captions and corrected the unit formatting to ensure accuracy and consistency (as suggested by Reviewers 1 and 2).

Our point-by-point responses to the reviews are below (reviewers' comments in black font, authors' responses in blue font). Line numbers referenced in the responses are based on the track-changes version of the revised manuscript. Some texts from the revised manuscript are shown in quotes where relevant.

Reviewer #1 (Remarks to the Author):

The manuscript presents an original and scientifically compelling framework linking western Eurasian snow cover anomalies to winter wildfire risk in Southern California. The observational–model synthesis is strong; however, the causal attribution could be strengthened by explicitly quantifying the contribution of competing drivers (e.g., ENSO, PDO, Arctic sea-ice anomalies) rather than describing them qualitatively. A sensitivity analysis or multi-model comparison section would help position the proposed mechanism relative to established teleconnections.

We thank the reviewer for the encouragement and the constructive suggestions. All comments have been carefully considered and addressed in the revised manuscript, and we hope that the revisions have adequately addressed the reviewer’s concerns.

Following the reviewer’s suggestion, we examined and compared the lead-lag correlations between the Southern California December–January Fire Weather Index (FWI) and other major climate indices, including El Niño–Southern Oscillation (ENSO), Pacific Decadal Oscillation (PDO), Pacific–North American pattern (PNA), West Pacific pattern (WP), North Atlantic Oscillation (NAO), Arctic Oscillation (AO), and the North Atlantic–Canadian Arctic sea-ice concentration (SIC) (Fig. R1). The SIC index is defined as the Arctic sea-ice concentration anomalies averaged over the North Atlantic–Canadian Arctic sector (150°W – 60°W , 55°N – 75°N ; informed by Fig. R2), and conventional definitions are used for the rest of climate indices. Among the indices considered, only the PNA and the AO indices exhibit moderate leading correlations with DJ Southern California FWI, with correlation coefficients of $r = -0.46$ for the October–November PNA and $r = 0.46$ for the October–November AO. This is not unexpected as AO and PNA are known to be two important climate modes that could influence North American climate variability. Though these correlations are weaker than that between November–December western Eurasian snow cover extent (SCE) and DJ Southern California FWI ($r = -0.59$).

Then, we constructed a new index to predict the December–January FWI in the Southern California using multivariate linear regression with SCE, ENSO, PDO, PNA, WP, NAO, AO, and SIC as predictors. Informed by our lead-lag analysis (Fig. R1), November–December average is used for SCE, and October–November averaged is used for other predictors. When SCE alone is used, it explains 35% of the variance of Southern California FWI index ($r = -0.59$). The explained variance increases to 44% or 42% when AO or PNA, respectively, is included in the multivariate linear regression. When all three factors (SCE, AO, PNA) are included, the explained variance further increases to 49%. When only AO and PNA are used, the explained variance drops to 35% without the contribution from SCE. Now, if all the factors mentioned above are included, the explained variance reaches 58%. These results indicate that, although various large-scale climate modes contribute to DJ fire weather conditions in Southern California, western Eurasian SCE emerges as a dominant contributor, followed by AO and PNA. We also examined the correlations among the different climate indices (Table R1). Moderate

correlations exist between some indices, and thus the explained variance by each predictor is not linearly additive. It is important to note that SCE exhibits very weak correlations with those indices.

Overall, incorporating western Eurasian SCE may help enhance the predictive skill for winter wildfire activity in California. These results have been added to the revised manuscript (Lines 389-401 in the track-changes version).

Figure R1. Correlation coefficients between the December–January (DJ) mean Southern California gridMET FWI and a suite of large-scale climate indices, including western Eurasian (10°E – 55°E & 48°N – 60°N) snow cover extent (SCE), El Niño–Southern Oscillation (ENSO), Pacific Decadal Oscillation (PDO), Pacific–North American pattern (PNA), West Pacific pattern (WP), North Atlantic Oscillation (NAO), Arctic Oscillation (AO) and North Atlantic–Canadian Arctic (210°E – 300°E , 55°N – 75°N) sea ice concentration (SIC) (a–h). In each panel, the green solid line shows the correlations between 3-month averaged climate index and DJ FWI; the red solid line shows the correlations between 2-month averaged climate index and DJ FWI; the blue dashed line shows the correlations between single-month climate index and DJ FWI. The horizontal axis represents the central time point of the climate index, with months to the left of the vertical black line indicating that the climate index leads the DJ mean FWI, and months to

the right indicating that climate index lags the DJ mean FWI. The vertical black line marks the zero-lag condition. The horizontal dashed line denotes the 95% significance threshold based on a two-sided Student's *t*-test.

Figure R2. Correlation maps between the December–January Southern California gridMET FWI and HadISST sea ice concentration (SIC) averaged over October–November, November–December, and December–January. Based on these results, we constructed an SIC index by area-averaging sea ice concentration over the North Atlantic–Canadian Arctic region (150°W – 60°W , 55°N – 75°N). This region was selected because it exhibits the most persistent and relatively stronger correlation with Southern California FWI among the Arctic sea ice regions, despite the overall correlations being weak.

Table R1. Correlation coefficients between different climate indices over the period 2000–2025. November–December average is used for SCE, and October–November averaged is used for other climate indices.

Corr.	SCE	ENSO	PDO	PNA	WP	NAO	AO
ENSO	0.21						
PDO	-0.06	0.49					
PNA	-0.33	-0.04	-0.01				
WP	-0.02	-0.43	-0.35	0.26			
NAO	0.08	-0.15	0.06	0.06	0.41		
AO	0.27	-0.24	-0.47	-0.21	0.44	0.55	
SIC	-0.06	0.09	0.35	0.07	0.06	0.46	0.46

Major comments

1. The CMIP6-based future risk interpretation remains somewhat speculative.

The manuscript currently implies future predictability based on the mechanistic link, but the robustness of this signal under warming—given snow regime shifts, fire–fuel nonlinearities, and model spread—is not fully assessed. A short uncertainty framing (even if exploratory) would make the conclusions more aligned with Nature Communications standards.

We thank the reviewer for this valuable suggestion. In response to this comment, we have added a paragraph in the Discussion section in the revised manuscript (Lines 422-440).

“The snow–fire teleconnection mechanism proposed in this study may have potential implications for assessing the future changes of wildfire activity in a warmer climate.

First, although this study is focused on interannual variability, the snow–fire teleconnection mechanism should also operate for externally forced climate changes. Whether the snow melt in western Eurasia under a warming climate will enhance winter wildfire activity in California awaits to be confirmed.

Second, as both SCE variability and atmospheric jets may alter under a warming climate with a rapid warming of the Arctic, the snow–atmosphere teleconnection and its impact on California wildfire activity may be non-stationary (Henderson et al., 2018). We attempt to explore this by investigating the snow–fire correlations in 30 pairs of historical and future simulations from 8 climate models in the CMIP6. Only a limited number of historical simulations exhibit statistically significant correlations, and none of them can reach the correlation seen in observations, although models reasonably simulate the magnitude of SCE variability itself (Fig. R3). This is consistent with previous studies showing that coupled climate models often struggle to internally capture key snow–atmosphere teleconnections (Furtado et al., 2015; Hardiman et al., 2008; Henderson et al., 2018). The model failure in reproducing the observed snow–fire linkage prevents us from drawing any solid conclusions on how the snow–fire teleconnection may possibly change under a warming climate.

Finally, fire–fuel nonlinearities may further increase the complexity of the future snow–fire relationship changes, as future warming can push fuel dryness across critical thresholds, triggering abrupt surges in fire activity (Zhao et al., 2025).

Therefore, future research along these lines is needed.”

Figure R3. a Standard deviation of November–December (ND) SCE in individual CMIP6 ensemble members during the historical period (2000–2025; blue dots) and under the SSP5-8.5 scenario (2075–2100; red dots). The ensemble-mean standard deviation, calculated across all 30 ensemble members, is highlighted with a larger dot. The blue square indicates the observed standard deviations of SCE from NOAA. **b** Correlation coefficients between the ND western Eurasian SCE index and the December–January (DJ) Southern California FWI index. The ensemble-mean correlation, calculated across all 30 ensemble members, is highlighted with a larger dot. The blue square indicates the observed correlation using NOAA SCE and gridMET FWI. In both panels, the dashed line marks the $p = 0.1$ significance threshold. Green vertical lines separate different ensemble members of the same model.

References

- Hardiman, S. C., Kushner, P. J. & Cohen, J. Investigating the ability of general circulation models to capture the effects of Eurasian snow cover on winter climate. *J. Geophys. Res. Atmos.* 113 (2008).
- Furtado, J. C., Cohen, J. L., Butler, A. H., Riddle, E. E. & Kumar, A. Eurasian snow cover variability and links to winter climate in the CMIP5 models. *Clim. Dyn.* 45, 2591–2605 (2015).
- Henderson, G. R., Peings, Y., Furtado, J. C. & Kushner, P. J. Snow–atmosphere coupling in the Northern Hemisphere. *Nat. Clim. Change* 8, 954–963 (2018).
- Zhao, H. et al. Future enhanced threshold effects of wildfire drivers could increase burned areas in northern mid-and high latitudes. *Comm. Earth Env.* 6, 224 (2025).

2. Validation of the fire weather signal would benefit from incorporating independent fire danger metrics beyond FWI and VPD (e.g., fuel moisture datasets, burned area residual analysis, or regional fire-weather classifications), particularly because winter fires are rare and observational record length is short.

We thank the reviewer for this valuable suggestion. We further examined independent fire-relevant metrics from the gridMET dataset during 2000-2025, including 1000-hour dead fuel moisture, 100-hour dead fuel moisture, and the Energy Release Component (ERC). The 1000-hour and 100-hour dead fuel moisture indices represent the moisture conditions of large and medium dead fuels, reflecting background and short-term fuel dryness, respectively. ERC integrates fuel moisture and cumulative drying conditions and serves as an indicator of potential wildfire intensity.

Linear regression analyses (Fig. R4) indicate that reduced western Eurasian snow cover extent is also associated with decreased dead fuel moisture and enhanced potential fire intensity over Southern California during December–January. Together, these changes reflect more fire-favorable fuel conditions and provide additional, independent support for the robustness of our main results. These results have been added to the revised manuscript (Lines 151-152).

Figure R4. Linear regression of December–January 1000-hour dead fuel moisture (%) (a), 100-hour dead fuel moisture (%) (b) and Energy Release Component (c) against the normalized NOAA November–December western Eurasian snow cover extent index (blue dashed line in Fig. 1a, multiplied by -1 to represent reduction) with ENSO removed. Stippling areas in panels (a–c) indicate responses significant at the 90% confidence level based on the two-sided Student’s t -test.

3. The authors should cite relevant work in which statistically derived fire weather indices have been compared against remotely sensed fire occurrence. In particular, Dhanurkar et al. (2024) demonstrated a strong spatiotemporal match between the Canadian Fire Weather Index and MODIS fire detections, which supports the methodological use of FWI as a proxy in regions or periods where burned area records are sparse. Including this strengthens the justification for using FWI in a low-fire-season context.

Thanks for bringing our attention to this important literature. Following the reviewer’s suggestion, we have added one sentence in Lines 91-94 and cited this paper as reference 39. This addition clarifies the methodological basis for using FWI as a proxy for wildfire activity, particularly in low-fire-season contexts where burned area records are sparse.

“Previous studies have demonstrated that the FWI corresponds well with observed wildfire events, including remotely sensed fire detections, supporting its use as a proxy for wildfire activity in regions or periods with limited burned area records^{23,39}.”

Minor comments

Figures 2 and 3 are dense and could benefit from clearer labeling and a more explicit visual guide to the causal sequence (e.g., schematic or pathway figure).

We thank the reviewer for this valuable suggestion. To improve clarity and guide the reader through the logical sequence of the processes, we added two labels in Fig. 2 indicating observed large-scale atmospheric teleconnection and local fire-weather condition associated with SCE reduction, and two labels in Fig. 3 indicating the corresponding simulated responses. We also added a conceptual schematic summarizing the proposed causal sequence (Fig. R5), which has been included in the revised manuscript as Fig. 5 (Lines 1059-1068).

Figure R5. Schematic diagram illustrating the proposed teleconnection linking reduced western Eurasian snow cover in early winter (November–December) to fire-favorable weather conditions in Southern California during December–January. Reduced snow cover over western Eurasia excites Rossby wave trains that establish large-scale atmospheric teleconnections, favoring the development of a persistent high-pressure system over the western U.S. with a typical wintertime western warming-eastern cooling dipole pattern across the continental U.S. This circulation

results in warmer and drier conditions, reduced fuel moisture, and enhanced offshore winds in Southern California, thereby increasing winter wildfire risk.

2. Please clarify whether detrending procedures were applied symmetrically across predictor and response variables, and whether the correlation strength is sensitive to the detrending method.

We thank the reviewer for raising this important point. In this study, the same preprocessing procedures were applied consistently to both the predictor and the response variables. Because neither the western Eurasian SCE nor the Southern California FWI exhibits a statistically significant long-term trend over the analysis period, detrending was not applied in our study. The correlations remain nearly unchanged when the indices are detrended, as described in Lines 135-137. We additionally summarize the correlation coefficients between different SCE indices and FWI indices with and without linear detrending in the table below (Table R2).

Table R2. Correlation coefficients between November–December (ND) western Eurasian SCE indices from NOAA, IMS, and MODIS, and December–January (DJ) Southern California FWI indices from gridMET and ERA5, over the period 2000–2025, with ENSO removed. The left and right columns show results without and with linear detrending, respectively.

	Without detrending	With detrending
gridMET FWI & NOAA SCE	-0.64 (p<0.01)	-0.64 (p<0.01)
gridMET FWI & IMS SCE	-0.59 (p<0.01)	-0.59 (p<0.01)
gridMET FWI & MODIS SCE	-0.54 (p<0.01)	-0.54 (p<0.01)
ERA5 FWI & NOAA SCE	-0.55 (p<0.01)	-0.54 (p<0.01)
ERA5 FWI & IMS SCE	-0.51 (p<0.01)	-0.51 (p<0.01)
ERA5 FWI & MODIS SCE	-0.45 (p=0.02)	-0.43 (p=0.03)

3. Some terminology (e.g., “bridge”, “pathway”) reads conceptual rather than mechanistic; consider aligning phrasing more closely with accepted atmospheric teleconnection nomenclature.

We agree with the reviewer on that terms such as “bridge” and “pathway” read conceptual rather than mechanistic. To address this concern, we have revised the manuscript to emphasize standard dynamical terms (e.g., replacing these expressions with “teleconnection” and “wave train”).

The term “bridge” is retained in the title in a descriptive sense, to denote the integrative linkage between remote cryosphere forcing and downstream wildfire-favorable conditions, while the underlying physical mechanisms are rigorously described using accepted atmospheric teleconnection terminology in the main text.

4. For reproducibility, the exact boundaries of the western Eurasian domain used in each analysis should be stated consistently in both main text and figures (some places define it only in captions).

We thank the reviewer for pointing this out. We have now consistently stated the exact latitude–longitude boundaries of the western Eurasian domain in the main text and the Methods section and ensured that the same definition is used throughout all figures and captions.

5. Minor typographical and formatting edits needed, particularly in multi-panel figure captions and units formatting.

We thank the reviewer for pointing this out. We have carefully reviewed and revised the formatting throughout the manuscript, with particular attention to the figure captions, and have ensured that all variables are clearly defined and consistently labeled with their corresponding units.

Reference

Dhanurkar, T., Budamala, V., & Bhowmik, R. D. (2024). Understanding the association between global forest fire products and hydrometeorological variables. *Science of The Total Environment*, 945, 173911.

We thank the reviewer for bringing this reference to our attention. As addressed above, it has been incorporated into the revised manuscript.

Reviewer #1 (Remarks on code availability):

I checked the codes which are mainly plotting codes.

We thank the reviewer for checking the code.

Reviewer #2 (Remarks to the Author):

This study links the record-breaking winter 2025 wildfires in southern California to anomalously low snow cover over western Europe and proposes a “snow–fire bridge” teleconnection: reduced European snow triggers a downstream Rossby-wave train that is enhanced by wave–mean-flow interaction over the North Pacific, thereby creating meteorological conditions conducive to fire in southern California. The work provides a novel, physically based global perspective on the remote drivers of such extreme events. The manuscript is generally well organized and the conclusions are robust, but the following points need clarification or additional analyses to reach Nature Communications’ standards.

We thank the reviewer for recognizing the novelty of our work and for providing many valuable and constructive comments. These comments have greatly helped us to improve the quality and clarity of the manuscript. We hope that our revisions have satisfactorily addressed the reviewer’s concerns.

Comments

1. This study used the FWI instead of burned area to investigate the remote drivers for the wildfires in South California. As shown in Fig. S3, the high FWI may provide a favorable condition for wildfires as in 2017 and 2024, but its impact is not necessarily inevitable, such as in 2011 and 2013. This should be clarified in the manuscript.

We thank the reviewer for raising this important point. We agree that a high FWI indicates favorable meteorological conditions for wildfire occurrence but does not necessarily guarantee that a large forest wildfire will occur, as fuel availability, ignition and fire suppression efforts also play critical roles (Abatzoglou & Williams 2016; Balch et al., 2017). We have clarified this point in the revised manuscript (Lines 96–99 in the track-changes version) by explicitly stating that FWI represents fire-conducive conditions rather than deterministic large fire burned area, other factors, such as ignition, fuel availability, and fire suppression efforts are also important.

References

- Abatzoglou, J. T. & Williams, A. P. Impact of anthropogenic climate change on wildfire across western US forests. *Proc. Nat. Acad. Sci. U.S.A.* 113, 11770–11775 (2016).
- Balch, J. K. et al. Human-started wildfires expand the fire niche across the United States. *Proc. Nat. Acad. Sci. U.S.A.* 114, 2946–2951 (2017).

2. Provide correlation coefficients (and significance levels) between western European snow anomalies and southern California winter FWI/burned area for 1920–2024 after excluding 2025, ideally with Monte-Carlo or bootstrap tests, to demonstrate that the teleconnection is not dominated by a single extreme year.

If we understood this question correctly, the reviewer was asking whether the correlation over the 2000–2024 period would be sensitive to the inclusion of the 2025 extreme event. To address the reviewer’s concern, we performed a bootstrap resampling analysis over the available observational period (2000–2025).

Correlations between western Eurasian SCE indices and Southern California winter FWI were calculated both including (N = 25) and excluding 2025 (N = 24), using gridMET and ERA5 FWI datasets. Statistical uncertainty was quantified using a nonparametric bootstrap approach, in which paired annual data were resampled with replacement 10,000 times to estimate 95% confidence intervals.

As shown in the figure below (Fig. R6), correlation coefficients change only slightly after excluding 2025, and the bootstrap confidence intervals remain entirely below zero across all snow datasets, indicating that this negative correlated relationship is not dominated by a single extreme year. We have included this analysis in the revised manuscript in Lines 137-138 as additional support for our findings. We thank the reviewer for this valuable suggestion.

Figure R6. Correlation coefficients between November–December western Eurasian SCE indices and December–January Southern California FWI indices. SCE is derived from three datasets (NOAA, IMS, and MODIS), and FWI is derived from (a) gridMET and (b) ERA5, with ENSO removed. Correlations are calculated both including 2025 (red; N = 25) and excluding 2025 (blue; N = 24). Black dots indicate the median correlation coefficients, and vertical bars denote the 95% confidence intervals estimated using a nonparametric bootstrap resampling approach with 10,000 iterations.

3. Snow-cover anomalies are surface-based, yet the key wave response is shown in the mid-to-upper troposphere. Please add a longitude–height cross-section of wave-activity flux (WAF or

EP-flux) along the wave path to demonstrate how the surface anomaly initiates upward-propagating Rossby waves. A brief dynamical discussion should also be included.

We thank the reviewer for this valuable suggestion. Following this comment, we have added a longitude–pressure cross-section of the WAF (F_x – F_z) averaged over 30°N – 60°N along the main wave path during December–January (Fig. R7). We chose the December–January average for the cross-section analysis because the atmospheric response is most robust during this mature stage, and it also coincides with the peak amplitude of snow cover extent anomalies. The cross-section clearly shows pronounced upward-propagating wave-activity flux over western Eurasia (0 – 60°E), indicating that the surface snow-cover extent anomalies act as a lower-boundary forcing that excites Rossby waves propagating upward into the mid-to-upper troposphere. Besides, we can also see how the surface anomaly initiates upward-propagating Rossby waves and how the Rossby waves intensifies above the North Pacific due to wave-mean flow interaction. Over 60° – 140°E , pronounced jet curvature degrades the waveguide, leading to weaker zonally averaged Rossby waves. This upward wave propagation over the snow forcing region provides a dynamical linkage between the surface snow anomalies and the downstream atmospheric circulation response over the North Pacific–North America sector. A brief discussion on these dynamical processes has been added to the revised manuscript in Lines 289-291.

Figure R7. A longitude–pressure cross-section of the WAF (F_x – F_z) averaged over 30°N – 60°N along the main wave path during December–January. For visualization purposes, the vertical component of the wave-activity flux (F_z) is multiplied by a factor of 100 to account for its smaller magnitude relative to the horizontal component.

4. The manuscript promotes Eurasian snow as a predictor. Quantify this: build a statistical model trained on 2001-2024 snow-cover indices and hindcast the 2025 event. Report cross-validated

skill scores and the amplitude of the predicted versus observed Southern California VPD/FWI. If skill is modest, state this clearly.

We thank the reviewer for this suggestion. To address this comment, we constructed a simple linear regression model using the November–December (ND) NOAA western Eurasian SCE index to predict Southern California December–January gridMET VPD/FWI index. The statistical model was trained on the western Eurasia ND SCE index over 2000–2024 and evaluated using leave-one-out cross-validation to assess out-of-sample predictive skill.

The snow-based model exhibits modest but statistically significant predictive skill (Leave-One-Out-Cross-Validation $r = 0.44$, $p = 0.03$ for FWI and $r=0.35$, $p=0.09$ for VPD). We then applied the trained model to hindcast the 2025 event using the NOAA-observed 2025 SCE index. The hindcast correctly predicts the sign of the 2025 standardized VPD/FWI anomaly but underestimates its magnitude. It can explain approximately 68% of the VPD anomaly and 40% of the FWI anomaly in 2025 from a statistical perspective. As suggested, we explicitly state that the predictive skill is modest. This analysis suggests that western Eurasian snow cover plays a contributing role in extreme wildfire-favorable weather conditions in Southern California and may provide additional predictive context rather than standalone forecasting skill. The corresponding description and results have been added to the revised manuscript in Lines 138–141.

5. The analysis assumes the relationship is stationary over the observational period. Given climate change and the rapid warming of the Arctic, the teleconnection pathway's strength and structure may be non-stationary. This should be discussed as a key uncertainty and limitation.

We thank the reviewer for this valuable comment and agree with this assessment.

The snow-fire teleconnection mechanism proposed in this study may have potential implications for assessing the future changes of wildfire activity in a warmer climate. But continued global and Arctic warming may alter the background circulation, including the strength and position of the polar and mid-latitude jet streams, which could in turn modify the snow–fire teleconnection identified in this study. This uncertainty and limitation is closely related to Reviewer #1's Comment 1 regarding future risk interpretation. In response to this comment, we have added a paragraph in the Discussion section in the revised manuscript (Lines 422–440).

“The snow-fire teleconnection mechanism proposed in this study may have potential implications for assessing the future changes of wildfire activity in a warmer climate.

First, although this study is focused on interannual variability, the snow-fire teleconnection mechanism should also operate for externally forced climate changes. Whether the snow melt in western Eurasia under a warming climate will enhance winter wildfire activity in California awaits to be confirmed.

Second, as both SCE variability and atmospheric jets may alter under a warming climate with a rapid warming of the Arctic, the snow–atmosphere teleconnection and its impact on California wildfire activity may be non-stationary (Henderson et al., 2018). We attempt to explore this by investigating the snow–fire correlations in 30 pairs of historical and future simulations from 8 climate models in the CMIP6. Only a limited number of historical simulations exhibit statistically significant correlations, and none of them can reach the correlation seen in observations, although models reasonably simulate the magnitude of SCE variability itself (Fig. R3). This is consistent with previous studies showing that coupled climate models often struggle to internally capture key snow–atmosphere teleconnections (Furtado et al., 2015; Hardiman et al., 2008; Henderson et al., 2018). The model failure in reproducing the observed snow–fire linkage prevents us from drawing any solid conclusions on how the snow–fire teleconnection may possibly change under a warming climate.

Finally, fire–fuel nonlinearities may further increase the complexity of the future snow–fire relationship changes, as future warming can push fuel dryness across critical thresholds, triggering abrupt surges in fire activity (Zhao et al., 2025).

Therefore, future research along these lines is needed.”

Thus, we agree that, from multiple perspectives, the conclusions of this study should be interpreted primarily in the context of the present climate, rather than as having direct applicability for future warming conditions.

References

- Hardiman, S. C., Kushner, P. J. & Cohen, J. Investigating the ability of general circulation models to capture the effects of Eurasian snow cover on winter climate. *J. Geophys. Res. Atmos.* 113 (2008).
- Furtado, J. C., Cohen, J. L., Butler, A. H., Riddle, E. E. & Kumar, A. Eurasian snow cover variability and links to winter climate in the CMIP5 models. *Clim. Dyn.* 45, 2591–2605 (2015).
- Henderson, G. R., Peings, Y., Furtado, J. C. & Kushner, P. J. Snow–atmosphere coupling in the Northern Hemisphere. *Nat. Clim. Change* 8, 954–963 (2018).
- Zhao, H. et al. Future enhanced threshold effects of wildfire drivers could increase burned areas in northern mid-and high latitudes. *Comm. Earth Env.* 6, 224 (2025).

6. European snow anomalies explain only ~one-third of the local vapor-pressure deficit (VPD). Please discuss: What drives the remaining VPD increment (e.g., local subsidence warming, advection of dry/warm air, SST anomalies)? In addition, why 2017 exhibited high FWI and large burned area despite weak snow anomalies, to delineate the boundary conditions of the snow–fire bridge.

We thank the reviewer for raising this important point. In fact, wildfire-favorable conditions are influenced by a combination of local processes and remote large-scale drivers. This comment is closely related to Reviewer #1's general concern regarding the role of competing climate drivers.

In response to this comment, we examined and compared the lead-lag correlations between the Southern California December–January Fire Weather Index (FWI) and other major climate indices, including El Niño–Southern Oscillation (ENSO), Pacific Decadal Oscillation (PDO), Pacific–North American pattern (PNA), West Pacific pattern (WP), North Atlantic Oscillation (NAO), Arctic Oscillation (AO), and the North Atlantic–Canadian Arctic sea-ice concentration (SIC) (Fig. R1). The SIC index is defined as the Arctic sea-ice concentration anomalies averaged over the North Atlantic–Canadian Arctic sector (150°W – 60°W , 55°N – 75°N ; informed by Fig. R2), and conventional definitions are used for the rest of climate indices. Among the indices considered, only the PNA and the AO indices exhibit moderate leading correlations with DJ Southern California FWI, with correlation coefficients of $r = -0.46$ for the October–November PNA and $r = 0.46$ for the October–November AO. This is not unexpected as AO and PNA are known to be two important climate modes that could influence North American climate variability. Though these correlations are weaker than that between November–December western Eurasian snow cover extent (SCE) and DJ Southern California FWI ($r = -0.59$).

Then, we constructed a new index to predict the December–January FWI in the Southern California using multivariate linear regression with SCE, ENSO, PDO, PNA, WP, NAO, AO, and SIC as predictors. Informed by our lead-lag analysis (Fig. R1), November–December average is used for SCE, and October–November average is used for other predictors. When SCE alone is used, it explains 35% of the variance of Southern California FWI index ($r = -0.59$). The explained variance increases to 44% or 42% when AO or PNA, respectively, is included in the multivariate linear regression. When all three factors (SCE, AO, PNA) are included, the explained variance further increases to 49%. When only AO and PNA are used, the explained variance drops to 35% without the contribution from SCE. Now, if all the factors mentioned above are included, the explained variance reaches 58%. These results indicate that, although various large-scale climate modes contribute to DJ fire weather conditions in Southern California, western Eurasian SCE emerges as a dominant contributor, followed by AO and PNA. We also examined the correlations among the different climate indices (Table R1). Moderate correlations exist between some indices, and thus the explained variance by each predictor is not linearly additive. It is important to note that SCE exhibits very weak correlations with those indices.

Thus, other internal variability like PNA and AO may drive the remaining VPD/FWI increment this year and may play a more dominant role in shaping wildfire-favorable conditions during the 2017 event. These results have been added to the revised manuscript (Lines 389–401).

7. The idealized snow experiments must be clearly detailed. Were they atmosphere-only (AGCM) runs with prescribed SSTs and snow? Or coupled? This has major implications for the interpretation.

The idealized snow experiments were conducted with the fully coupled CESM2.1 model rather than an atmosphere-only (AGCM) model. While the model is fully coupled, our interpretation focuses on the atmospheric circulation response to the imposed western Eurasian snow cover anomalies, because the associated North Pacific SST response is quite weak ($< 0.1\text{ }^{\circ}\text{C}$) (Fig. R8a), and therefore unlikely to exert substantial feedback on the winter atmospheric circulation. Thus, in this study, the dominant pathway involves eastward-propagating Rossby waves and wave–mean flow interactions over the North Pacific, rather than being driven primarily by North Pacific ocean–atmosphere coupling. Besides, the response of Arctic SIC is also negligible (Fig. R8b). We have clarified the experimental design and the corresponding interpretation in detail in the revised manuscript (Lines 561-567).

Figure R8. Simulated December-January mean (a) SST ($^{\circ}\text{C}$) and (b) SIC (unitless) response.

8. Unit is missing in Fig. S8

We have added this unit (W/m^2). Thank you for pointing it out.

Responses to the reviews of the manuscript

“A Snow-Fire Bridge Mechanism for 2025 Southern California Winter Wildfire”

Shizuo Liu, Shineng Hu, Richard Seager

We sincerely thank the reviewers for dedicating their time to evaluating our work and for providing both encouraging feedback and constructive suggestions. Their insightful comments have greatly improved the clarity and rigor of our study. We have carefully addressed the reviewers' suggestions in the revised version.

Reviewer #1 (Remarks to the Author):

I have carefully examined the revised manuscript and the authors' detailed point-by-point responses. I appreciate the substantial effort invested in addressing the concerns raised during the previous review. The additional analyses, clarifications, and expanded discussion have significantly strengthened the robustness, transparency, and overall presentation of the study.

In particular, the inclusion of bootstrap testing, independent fire danger metrics, the statistical hindcast experiment, and the expanded discussion on uncertainties and non-stationarity have addressed my main concerns. The revisions are thorough and appropriate.

I am satisfied with the revision and support the manuscript for publication in its current form.

We thank the reviewer for the careful reassessment and for confirming that the previous concerns have been addressed.

Reviewer #2 (Remarks to the Author):

My concerns have been satisfactorily addressed. I have no further comments.

We are glad that the reviewer's concerns have now been satisfactorily addressed.

Reviewer #2 (Remarks on code availability):

I was able to run the codes and reproduce the results. To make this more useful for the community, it would be helpful if the author could package the main logic into a reusable function.

We thank the reviewer for this helpful suggestion. We have provided the main analysis code in the Code Ocean capsule and refactored the dynamical diagnostics component into reusable callable functions. Usage instructions have been added to the README to facilitate broader use.